



# Weakening of springtime Arctic ozone depletion with climate change

Marina Friedel[1], Gabriel Chiodo[1], Timofei Sukhodolov[2], James Keeble[3,4], Thomas Peter[1],
Svenja Seeber[1], Andrea Stenke[1,5,6], Hideharu Akiyoshi[7], Eugene Rozanov[2], David Plummer[8],
Patrick Jöckel[9], Guang Zeng[10], Olaf Morgenstern[10], and Béatrice Josse[11]

[1]Institute for Atmospheric and Climate Sciences, ETH Zurich, Zurich, Switzerland
[2]Physikalisch-Meteorologisches Observatorium Davos/World Radiation Center, Davos, Switzerland
[3]Yusuf Hamied Department of Chemistry, University of Cambridge, Cambridge, UK
[4]National Centre for Atmospheric Science (NCAS), University of Cambridge, Cambridge, UK
[5]ETH Zürich, Institute of Biogeochemistry and Pollutant Dynamics, Zürich, Switzerland
[6]Eawag, Swiss Federal Institute of Aquatic Science and Technology, Dübendorf, Switzerland
[7]National Institute for Environmental Studies, Tsukuba, Japan
[8]Climate Research Division, Environment and Climate Change Canada, Montreal, Canada
[9]Deutsches Zentrum für Luft- und Raumfahrt (DLR), Institut für Physik der Atmosphäre, Oberpfaffenhofen, Germany
[10]National Institute of Water and Atmospheric Research (NIWA), Wellington, New Zealand
[11]Centre National de Recherches Météorologiques, Université de Toulouse, Météo-France, CNRS, Toulouse, France

**Correspondence:** Marina Friedel (marina.friedel@env.ethz.ch)

**Abstract.** In the Arctic stratosphere, the combination of chemical ozone depletion by halogenated ozone-depleting substances (hODSs) and dynamic fluctuations can lead to severe ozone minima. These Arctic ozone minima are of great societal concern due to their health and climate impacts. Owing to the success of the Montreal Protocol, hODSs in the stratosphere are gradually declining, resulting in a recovery of the ozone layer. On the other hand, continued greenhouse gas (GHG) emissions cool the

stratosphere, possibly enhancing the formation of polar stratospheric clouds (PSCs) and, thus, enabling more efficient chemical ozone destruction. Other processes, such as the acceleration of the Brewer-Dobson circulation, also affect stratospheric temperatures, further complicating the picture. Therefore, it is currently unclear whether major Arctic ozone minima will still occur at the end of the 21st century despite decreasing hODSs. We have examined this question for different emission pathways using simulations conducted within the Chemistry-Climate Model Initiative (CCMI-1 and CCMI-2022) and find large differences

in the models' ability to simulate the magnitude of ozone minima in the present-day climate. Models with a generally too cold polar stratosphere ("cold bias") produce pronounced ozone minima under present-day climate conditions, because they simulate more PSCs and, thus, high concentrations of active chlorine species (ClOx). These models predict the largest decrease in ozone minima in the future. Conversely, models with a warm polar stratosphere ("warm bias") have the smallest sensitivity of ozone minima to future changes in hODS and GHG concentrations. As a result, the scatter among models in the magnitude

of Arctic spring ozone minima will decrease in the future. Overall, these results suggest that Arctic ozone minima will become weaker over the next decades, largely due to the decline in hODS abundances. We note that none of the models analysed here project a notable increase of ozone minima in the future. Stratospheric cooling caused by increasing GHG concentrations is expected to play a secondary role, as its effect in the Arctic stratosphere is weakened by opposing radiative and dynamical mechanisms.





## 1   Introduction

The springtime Antarctic ozone hole is driven by chemical ozone destruction linked to the abundance of anthropogenic halogen containing ozone depleting substances (hODSs) in a strong, cold polar vortex. Dynamical variability plays an important role in modulating this chemical depletion on interannual timescales. At sufficiently cold temperatures, chlorine and bromine is activated through heterogeneous reactions occurring on polar stratospheric clouds (PSCs), leading to ozone depletion via

catalytic cycles (Solomon, 1999). With the success of the Montreal Protocol and its amendments (MPA) in controlling the emissions of chlorine and bromine containing substances, chemical ozone depletion is expected to decline, and the Antarctic ozone hole is expected to recover in the second half of the 21st century (Dhomse et al., 2018; Amos et al., 2020; WMO, 2022). Several studies suggest that early signs of recovery can already be detected (Várai et al., 2015; Solomon et al., 2016; Kuttippurath and Nair, 2017; Chipperfield et al., 2017).

Large seasonal ozone loss, albeit less frequently, also occurs in sufficiently cold Arctic springs (Manney et al., 2011, 2020). Despite being relatively infrequent, Arctic ozone minima have a great societal relevance because of their potential impacts on health and climate (Norval et al., 2011; Friedel et al., 2022a). Large interannual variability in Arctic ozone has so far masked potential signs of recovery in the Northern Hemisphere (NH) (WMO, 2018). Moreover, ongoing emission of greenhouse gases (GHGs) radiatively cool the stratosphere (Eyring et al., 2007; Pommereau et al., 2018), potentially increasing the abundance of

PSCs, leading to more effective ozone depletion in cold boreal springs — despite decreasing hODSs. In this context, it has been estimated that 1 K of polar stratospheric cooling could offset a reduction of hODSs of 10% (Sinnhuber et al., 2011). In addition to temperature changes, an increase in stratospheric water vapour due to GHG-induced changes in tropopause temperature could favour the formation of PSCs (Keeble et al., 2021; von der Gathen et al., 2021).

An increase in PSC volume in particularly cold winters since the 1980s, due to stratospheric cooling by GHGs, has previously

been suggested as a driver of recent Arctic ozone depletion (Shindell et al., 1998; Rex et al., 2004, 2006; Tilmes et al., 2006; von der Gathen et al., 2021). However, trends in Arctic temperature and PSC volume are difficult to detect in observations due to the short observational record and large interannual variability, and significant trends in PSC abundance in the observational record have been called into question (Hitchcock et al., 2009; Rieder and Polvani, 2013). Furthermore, past declines in Arctic stratospheric temperature cannot be attributed with confidence to an increase in GHGs (Rex et al., 2004; Rieder et al., 2014).

Rather, hODSs have been suggested to be the cause of past stratospheric temperature changes of particularly cold Arctic springs via ozone depletion (Hitchcock et al., 2009; Rieder et al., 2014).

For the current (21st) century, it has been suggested that the continuous rise of GHG concentrations might play an important role in the recovery of the ozone layer and could be responsible for a delay in Arctic ozone return dates (Pommereau et al., 2018). However, there is no consensus on the impact of increasing GHGs on springtime Arctic stratospheric temperature and

associated effects on Arctic ozone depletion events. While some studies with chemistry-climate models (CCMs) do not find robust evidence of cooling trends in the Arctic over the next century (Eyring et al., 2007; Rieder and Polvani, 2013; Langematz et al., 2014; Bohlinger et al., 2014), other CCM model simulations show the potential of large ozone depletion events, even beyond 2060 (Bednarz et al., 2016; Akiyoshi et al., 2023). In addition, CMIP6 models project an increase in PSC formation



potential by the end of the century in high emission scenarios, which may lead to occasional strong depletion of Arctic ozone
(von der Gathen et al., 2021).

The large uncertainty in future Arctic ozone depletion across models is the result of different model sensitivities to GHG and
hODS forcings, especially with respect to lower-stratospheric transport and dynamical responses (Morgenstern et al., 2018),
leading to a large inter-model spread in temperature and ozone trends. In addition, stratospheric temperature trends depend
on the GHG emission scenario studied, posing an additional source of uncertainty (von der Gathen et al., 2021). The study at
hand aims at shedding new light on the evolution of Arctic ozone minima in future climate by comparing the ozone evolution
across different CCMs and GHG emission scenarios, while identifying reasons for model discrepancies. By linking the model
spread in future ozone trends to differences in model climatologies, comparison with observations allows us to identify the
likely evolution of future Arctic ozone minima.

## 2   Materials and Methods

With the two CCMs SOCOL-MPIOM and WACCM version 4 we perform both transient simulations of the 21st century for
different emission scenarios, as well as timeslice simulations of the years 2000 and 2075. We further compare the results
with corresponding simulations of the Chemistry-Climate Model Initiative (CCMI), CCMI-1 and CCMI-2022, as well as the
reanalysis dataset MERRA2.

### 2.1   Chemistry-Climate Modelling

WACCM, the Whole Atmosphere Community Climate Model, is the atmospheric component of the NCAR Community Earth
System Model version 1 (CESM1.2.2). WACCM has a horizontal resolution of 1.9° in latitude and 2.5° in longitude (Marsh
et al., 2013) and is coupled to interactive ocean and sea ice components (Danabasoglu et al., 2012; Holland et al., 2012).
Ozone concentrations are calculated interactively including a total of 59 species (Marsh et al., 2013). With its well resolved
stratosphere and high model top ($5.1 \times 10^{-6}$ hPa) on 66 vertical levels (Marsh et al., 2013), WACCM has been documented
to capture stratospheric trends and variability reasonably well (Haase and Matthes, 2019; Rieder et al., 2019; Oehrlein et al.,
2020).

SOCOL, SOlar Climate Ozone Links, version 3 is based on the general circulation model MA-ECHAM5, which is inter-
actively coupled to the chemistry transport model MEZON (Model for Evaluation of oZONe trends (Egorova et al., 2003)).
The model version SOCOL-MPIOM is additionally coupled to the ocean-sea-ice model MPIOM (Stenke et al., 2013; Muthers
et al., 2014). SOCOL-MPIOM has a model top at 0.01 hPa and 39 vertical levels and a horizontal resolution of 3.75° × 3.75°
(Stenke et al., 2013). Ozone is calculated interactively based on a set of 140 gas-phase, 46 photolysis and 16 heterogeneous
reactions involving 41 species. Like WACCM, SOCOL-MPIOM captures stratospheric variability reasonably well (Muthers
et al., 2014).

To gain understanding of the dependency of stratospheric ozone trends and variability in the Arctic on the GHG loading, we
compare simulations of high and low emission scenarios over the 21st century (2005-2099). For the high emission scenario,



GHGs follow the RCP8.5 pathway, whereas the low emission scenario is based on the RCP2.6 pathway (Meinshausen et al., 2011). hODSs follow the A1 scenario according to WMO (2014). In addition to the transient simulations, we perform timeslice simulations of the early (year 2000) and late (year 2075) 21st century with fixed, seasonally varying GHGs and hODSs of the respective year. For simulations of the year 2075, boundary conditions follow the RCP8.5 pathway. In these timeslice simulations, covering 200 years each, trends in stratospheric ozone and climate are omitted, allowing to robustly assess long-term changes in the variability due to changes in GHGs and hODS levels.

## 2.2 CCMI simulations

We compare our model results with sensitivity simulations of high (SEN-C2-RCP85) and low (SEN-C2-RCP26) emission scenarios conducted for phase 1 of CCMI (CCMI-1). GHG emissions of those simulations follow the RCP8.5 and RCP2.6 pathway (Meinshausen et al., 2011), respectively, with hODSs following the A1 scenario (WMO, 2011) . The high and low emission pathways therefore only differ in their assumptions on future GHG emissions, while hODSs are equal for both scenarios. Model simulations usually cover the period 2000-2100. We analyse all models that performed the SEN-C2-RCP85 simulation, namely SOCOL CCMI, WACCM CCMI, CCSRNIES-MIROC3.2, CMAM, EMAC-L47MA, IPSL, ULAQ-CCM and UMSLIMCAT, with a subset of them also performing the SEN-C2-RCP26 simulation. Description of CCMI-1 models is presented by Morgenstern et al. (2017).

To investigate the robustness of the evolution of ozone minima in CCMI models, we perform out-of-sample testing with CCMI-2022 simulations (SPARC, 2021). The moderate emission scenario of CCMI-2022 simulations (REF-D2) follows the SSP2-4.5 pathway (Meinshausen et al., 2020), and hODSs follow the WMO (2018) baseline scenario. We analyse all models that performed the REF-D2 scenario, namely CNRM-MOCAGE, NIWA-UKCA2, CCSRNIES-MIROC3.2, CMAM and EMAC-CCMI2. Model simulations were generally conducted for the time period 1960-2100. Here, we analyse the period 2000-2100. In addition to the CCMI-2022 simulations, we perform simulations with SOCOLv4 (Sukhodolov et al., 2021) following the CCMI-2022 REF-D2 boundary conditions. SOCOLv4 is an updated version of SOCOL-MPIOM based on the Earth system model MPI-ESM1.2 (Mauritsen et al., 2019), and is additionally coupled to the sulfate aerosol microphysical model AER (Weisenstein et al., 1997). An overview of all model simulations and available ensemble members that were analysed in this study can be found in Table 1. For all models, we use the model output for Arctic mean ozone, temperature and, where available, active chlorine species ($ClOx = ClO + Cl + 2\ Cl_2O_2$) interpolated to pressure levels.

## 2.3 Reanalysis

Model results for the early 21st century are compared to the Modern-Era Retrospective Analysis for Research and Applications version 2 (MERRA2), from 1980 to 2020 (Gelaro et al., 2017). MERRA2 has a horizontal resolution of $0.5° \times 0.625°$ and 72 vertical levels with a model top at 0.01 hPa, and we use 6-hourly instantaneous data output, which is then converted into monthly and springtime averages. MERRA2 has been shown to agree well with satellite and ozone sonde data regarding stratospheric ozone variability (Wargan et al., 2017; Davis et al., 2017; Bahramvash Shams et al., 2022).



**Table 1.** Model Experiments analysed in this study.

| Project | Model | Years | Scenarios | Ensemble members |
|---------|-------|-------|-----------|------------------|
| This study | SOCOL-MPIOM | 200 | timeslice year 2000 | 1 |
| | WACCM4 | 200 | timeslice year 2000 | 1 |
| | SOCOL-MPIOM | 196 | timeslice year 2075 | 1 |
| | WACCM4 | 200 | timeslice year 2075 | 1 |
| | SOCOL-MPIOM | 2003-2099 | RCP8.5, RCP2.6 | 5,3 |
| | WACCM4 | 2005-2099 | RCP8.5, RCP2.6 | 5,2 |
| CCMI-1 | SOCOL CCMI | 2000-2099 | RCP8.5 | 1 |
| | WACCM CCMI | 2005-2099 | RCP8.5 | 1 |
| | CCSRNIES-MIROC3.2 | 2000-2100 | RCP8.5, RCP2.6 | 1,1 |
| | CMAM | 2001-2100 | RCP8.5, RCP2.6 | 1,1 |
| | EMAC-L47MA | 2000-2099 | RCP8.5 | 1 |
| | IPSL | 2000-2094 | RCP8.5, RCP2.6 | 2,1 |
| | ULAQ-CCM | 2000-2100 | RCP8.5, RCP2.6 | 1,1 |
| | UMSLIMCAT | 2000-2099 | RCP8.5 | 1 |
| CCMI-2022 | CNRM-MOCAGE | 2000-2099 | SSP2-4.5 | 1 |
| | NIWA-UKCA2 | 2000-2100 | SSP2-4.5 | 3 |
| | CCSRNIES-MIROC3.2 | 2000-2100 | SSP2-4.5 | 1 |
| | CMAM | 2000-2100 | SSP2-4.5 | 3 |
| | EMAC-CCMI2 | 2000-2099 | SSP2-4.5 | 3 |
| | SOCOLv4 | 2000-2099 | SSP2-4.5 | 3 |

## 2.4 Analysis Methods

Our analysis focuses on boreal spring, which we define as March – April averages, where ozone anomalies maximize in the
models. Unless stated otherwise, the results below show averages over the polar cap, defined as 60-90°N, applying latitudinal
weighting according to the cosine of latitude. Lower stratospheric ozone is defined as partial ozone column between 30 and 70
hPa in DU (Friedel et al., 2022a). Ozone distributions shown contain all ensemble members of a simulation (without averag-
ing). The mean magnitude of the ozone minima is defined as the lowest 20th percentile of springtime ozone anomalies for both
timeslice and transient (scenario) simulations. For timeslice simulations, we thus average over the 40 lowest springtime ozone
anomalies to derive the mean ozone minima strength. For transient simulations, we calculate the mean magnitude of ozone
minima in a 25-year running window. For each window, the anomaly in each spring is calculated relative to the springtime
climatology of that window, and the mean ozone minima is then calculated by averaging over the lowest 20th percentile of
ozone anomalies (i.e. 5 lowest springtime ozone values out of 25 springs). Uncertainty in the magnitude of the ozone minima is





defined as the sensitivity to including one more (6 out of 25) and one less (4 out of 25) springs into the calculation. For ensem-
ble simulations with multiple members, this analysis is conducted for each ensemble member individually before averaging.
Similarly, the temperature of particularly cold winters is defined as the lowest 20th percentile of springtime temperature at 50
hPa in a 25-year running window, and the uncertainty of this variable is calculated by including one more or less spring in the
calculation. For climatologies, uncertainties are given as standard deviations from the mean. For simulations containing mul-
tiple ensemble members, climatologies are calculated as ensemble means, and uncertainties are given as the root mean square
of the individual members' uncertainties. Trends are generally estimated by linear least squares regressions of the respective
ensemble mean, and uncertainty in trends is given by the standard error of the slope. Significance of trends is estimated by
p-values calculated using a Wald Test (Fahrmeir et al., 2013).

## 3 Results

### 3.1 Evolution of Arctic ozone minima in CCMI models

To identify the spread in simulated Arctic ozone and the evolution thereof across CCMs, we start by analysing springtime Arctic
ozone anomalies in the early (2005 – 2029) and late (2070 – 2094) 21st century. Anomalies are defined as deviations from the
climatology of the respective time period. Figure 1 shows distributions of Arctic lower stratospheric ozone anomalies for the
high emissions scenario (RCP8.5) simulated with WACCM, SOCOL-MPIOM and CCMI-1 models. In the following, we will
concentrate on negative ozone anomalies. For the early time period (Fig. 1 a), differences in the magnitude of ozone anomalies
across models are striking; while some models simulate negative ozone anomalies larger than -40 DU for the springtime mean
lower stratospheric ozone column, other models barely show anomalies exceeding -5 DU. In the reanalysis product MERRA2,
which covers recent past to present-day climate (1980–2020), negative ozone anomalies reach up to -40 DU. Thus, most of the
CCMs (7 out of 10) do not reproduce the most extreme negative ozone anomalies under current climatic conditions compared
to reanalysis. In comparison, by the end of the century, CCMs provide a more coherent picture; almost all models simulate
negative ozone anomalies of a maximum of -20 DU. Thus, according to CCM simulations, extreme ozone loss beyond -20 DU
is unlikely past 2070 in the high emission scenario. Large ozone variability and extreme ozone loss comparable to MERRA2
is not projected by any model at the end of the century.

Further to changes in ozone anomalies, it is important to note that the mean ozone climatology is changing across the two
time periods considered. Under the high emission scenario RCP8.5, Arctic stratospheric ozone is expected to increase from
136 DU to 153 DU on a multimodel mean over the course of the 21st century (see Fig. A1) due to the combined impact of
a range of processes: the decrease in hODSs, a strengthening of the Brewer-Dobson circulation (BDC) by GHGs (i.e. larger
transport of ozone from the tropics to the poles) (Butchart, 2014), stratospheric cooling by GHGs in the tropical and mid-
latitude region (which slows down ozone depletion there), and increasing methane concentrations (Revell et al., 2012). This
projection of Arctic ozone recovery of 17 DU from 2000 until 2100 is consistent with what has been reported previously for
the lower stratosphere in CCMI-1 models (Dhomse et al., 2018). The rise in mean ozone suggests that even during the most





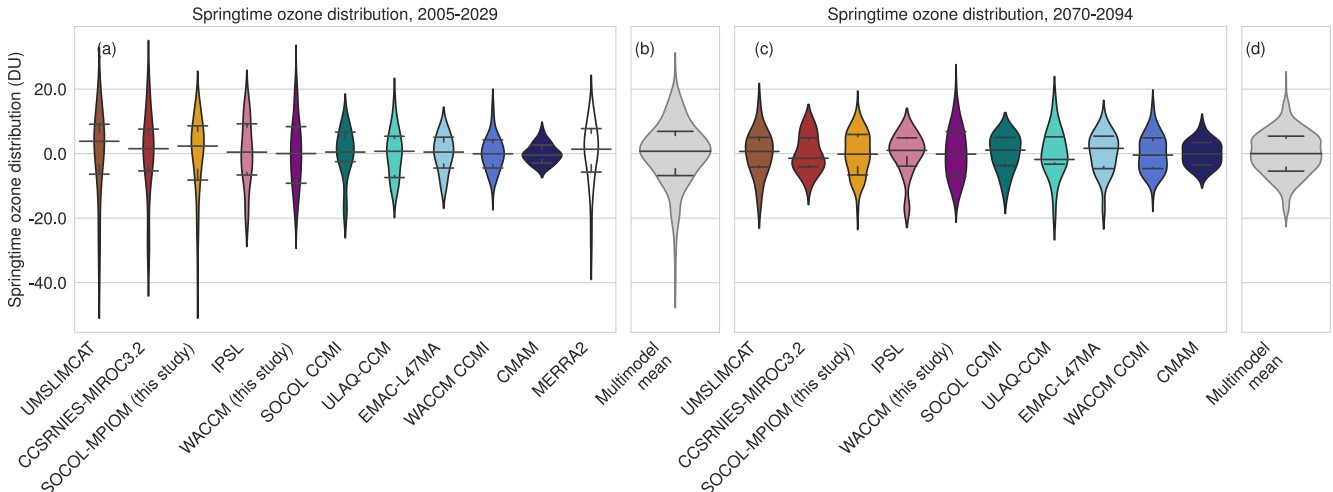

**Figure 1.** Probability density distributions calculated based on kernel density estimation of stratospheric partial ozone column (30-70 hPa) in spring (March-April) normalized by the mean ozone of the respective time period in the early (2005-2029, a, b) and late (2070-2094, c, d) 21st century in CCMI-1 models for RCP8.5. Whiskers show 20th and 80th percentile of the distributions. The multi-model mean shows the distributions over all models (excluding MERRA2) containing each ensemble member with equal weight. Similar Figs. for RCP2.6 and CCMI-2022 can be found in the supplementary material (Figs. A2 and A3).

severe projected negative anomalies of -20 DU by the end of the century, there will be hardly less ozone than on the current mean.

In the following, we will focus on the most extreme negative ozone anomalies, defined as the lower 20th percentile of springtime ozone anomalies and referred to as "ozone minima". To examine changes in ozone minima over time, we calculate
165   the ozone minima in a 25-year running window (i.e. average over the 5 most extreme negative ozone anomalies out of 25 springs). Figure 2 shows the development of the ozone minima over time for all model simulations and different GHG emission pathways: RCP8.5 (Fig. 2 a), RCP2.6 (Fig. 2 c) and SSP2-4.5 (Fig. 2 e). Again, there is a large scatter across the models in terms of the strength of the simulated ozone minima at the beginning of the century, ranging from -22.5 DU to -3 DU. Over time, the model spread decreases, and by the end of the century it ranges from -10 DU and -3 DU. The uncertainty in the
170   magnitude of ozone minima therefore reduces over the course of the century. Models that simulate large ozone minima under current conditions (2005 – 2029) also show the largest ozone minima under future conditions (e.g. UMSLIMCAT), but the magnitude of these future minima is significantly smaller. In contrast, models with small ozone minima under present day conditions (e.g. CMAM) show hardly any change in the magnitude of these minima under future conditions.

The future decline in the magnitude of ozone minima is related to the magnitude of the Arctic ozone depletion under current
175   conditions (see negative correlations of -0.84 to -0.96 in Fig. 2 b, d, f). The development of ozone minima is thereby strongly correlated with the initial strength of the ozone minima in the respective model. Consequently, models with large ozone minima at the beginning of the century in general show larger trends towards less pronounced ozone minima in the future, whereas





models with small ozone minima show no trends at all. Linear regression of the trend in ozone minima to their initial magnitude shows that this relationship is independent of the emission scenario, i.e. the regression slope is the same across all scenarios considered ($s = -0.1$). Given this strong correlation, the large inter-model spread can be used to constrain projections of future ozone minima by observations. To this end, we compare the simulated ozone minima in present-day climate with ozone minima observed during the past 40 years (1980 – 2020) in MERRA2 (black line in Figs. 2 b, d, f) and find that the models that most realistically reproduce current ozone minima project a decrease in extreme negative ozone anomalies of about 1 DU decade$^{-1}$ (see intersection of regression line and reanalysis). Hence, using past ozone minima in reanalysis as an emergent constraint suggests that extreme negative ozone anomalies will likely be 8 – 10 DU (and therefore around 50%) less severe by the end of the 21st century. It is to be noted that an emergent constraint analysis can only be a useful tool for reducing uncertainty in future projections if it (1) survives out-of-sample testing, and (2) exhibits a physical mechanism underlying the strong statistical relationship (Hall et al., 2019; Simpson et al., 2021). While the former is fulfilled by the independence of the results from emission pathways and model sets, the latter will be discussed below.

A more precise method for estimating the likely evolution of ozone minima are weighted model means. Here, we calculate weights for each individual model based on their ability to reproduce the present-day ozone minima ("performance") and their interdependence (i.e. the family-relationship of some models) (Knutti et al., 2017; Amos et al., 2020; Morgenstern et al., 2017). The model weights are then used to calculate a weighted arithmetic mean of the trend in ozone minima. Resulting model weights and a more detailed description of the methodology can be found in the appendix (section A2). Weighted model means suggest a reduction of the magnitude of ozone minima of 1.0 (RCP8.5), 1.1 (RCP2.6) and 1.0 (SSP2-4.5) DU decade$^{-1}$. Thus, weighted model means are largely scenario-independent and agree well with estimates of the emergent-constraint approach.

Since many models only simulated one single ensemble member for future projections (see Table 1) and the sample used for calculating the ozone minima is small (5 springs per running window width), we test the sensitivity of our results to sample size using 200-year time-slice simulations performed with the CCMs WACCM and SOCOL-MPIOM. By using fixed boundary conditions of the years 2000 and 2075, we omit any trends in ozone or temperature due to changes in hODSs and GHGs. Consistent with the previous definition, we again select the 20th percentile of most extreme negative ozone anomalies (40 out of 200) to define the magnitude of the mean ozone minima in timeslice simulations. The ozone minima in these simulations, shown as circles in Fig. 2 a, agree very well with the transient simulations, suggesting that the smaller sample size in transient simulations is sufficient to derive robust results (see also Fig. A6).

## 3.2 The source of model differences in current climate

We have shown that model dispersion in simulated ozone can be useful in constraining the evolution of ozone minima. However, the question arises as to why CCMs show large differences in the magnitude of ozone minima under current climatic conditions in the first place. Arctic ozone minima are caused by both chemical ozone depletion and dynamical variability (Tegtmeier et al., 2008); usually, ozone minima are preceded by a reduced wave activity, and consequently a strengthening and cooling of the polar vortex as well as a reduction of the BDC. Dynamical resupply of ozone to the polar region is therefore reduced, and cold temperatures allow for the formation of PSCs and chlorine activation, consequently leading to ozone depletion. The amount of



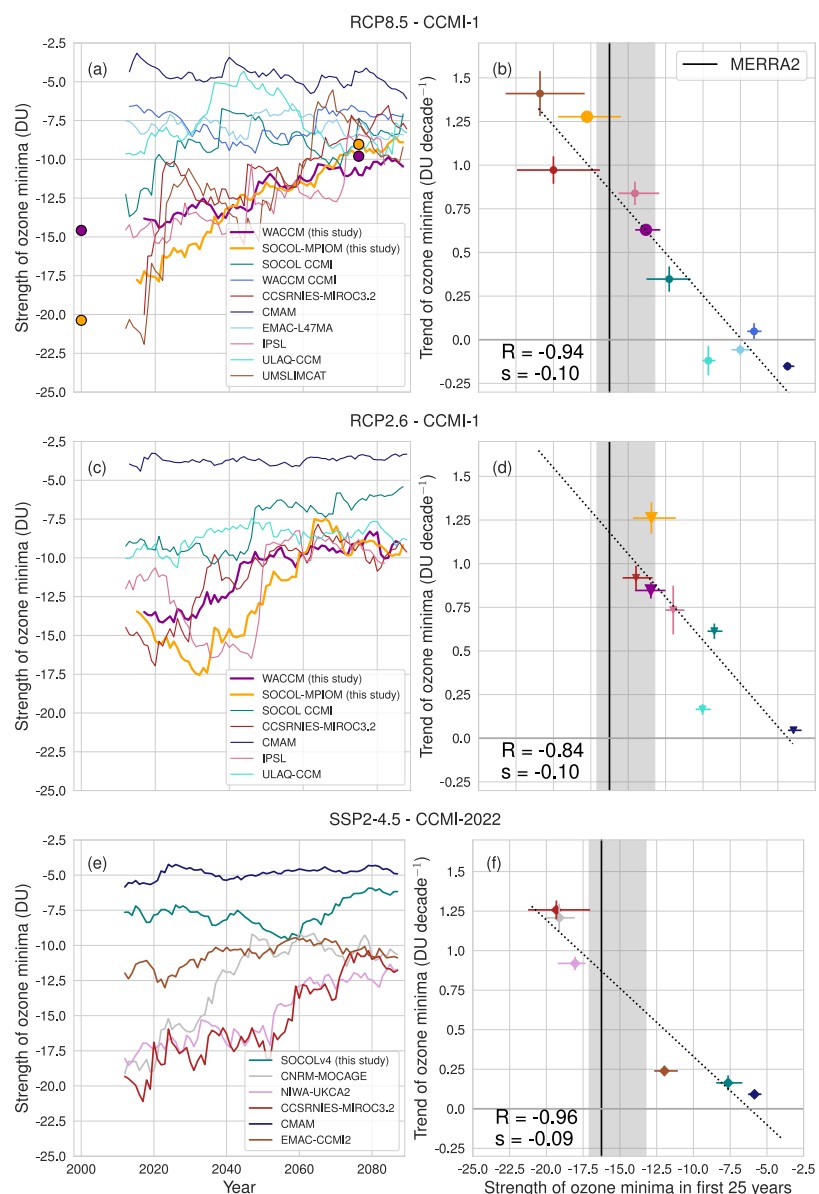

**Figure 2.** Development of the strength of Arctic ozone minima in CCMI-1 models under RCP8.5 (a) and RCP2.6 (c), as well as for CCMI-2022 models under SSP2-4.5 (e). Correlation of the trend in ozone minima strength and the strength of the ozone minima in the first 25 years defined by the mean ozone anomaly of the 20th percentile (5 out of 25 strongest ozone minima in running window, normalized by ozone climatology of running window). The mean strength of ozone minima in MERRA2 from 1980-2020 is given by the solid black line. The grey shading shows the uncertainty of the MERRA2 ozone minima strength, as explained in the methods section. Circles in (a) show the ozone minima strength of the WACCM and SOCOL-MPIOM timeslice simulations for the years 2000 and 2075, respectively.





ozone depleted thereby depends on the amount of active chlorine species (ClOx) in the stratosphere. In Fig. 3 a, we show that
the strength of ozone minima strongly correlates with the stratospheric Arctic mean ClOx concentrations (at 50 hPa) across
models in the beginning of the 21st century. Hence, the differences in the magnitude of ozone minima in different models are

attributable to differences in chemical ozone destruction by ClOx. The amount of activated chlorine in the stratosphere itself
depends on the volume of the PSCs, which in turn is strongly temperature-dependent. We find a large spread (around 10 K) in
the mean Arctic stratospheric temperature across models in winter/spring (January – April). This model scatter is consistent
with what has been reported previously by Morgenstern et al. (2022) for some CMIP6 and CCMI-2022 models. Reasons for
such differences in the models' mean stratospheric springtime temperature are likely linked to differences in the large-scale

circulation, such as the BDC, or the lifetime and shape of the polar vortex. Here, we complement the analysis by Morgenstern
et al. (2022) connecting those temperature biases to ClOx concentrations and finally the magnitude of ozone minima. When
linking the mean polar cap temperature at 50 hPa to stratospheric ClOx concentrations at the same altitude in the CCMs, we find
a linear relationship with a correlation coefficient of $R = -0.69$ (see Fig. 3 b). Thus, differences in ClOx concentrations among
models can to a large part be attributed to temperature biases. Models which show a warm temperature bias (e.g. CMAM, dark

blue markers) under present-day conditions compared to MERRA2 (vertical solid black line in Fig. 3 b) in general have small
ClOx concentrations (Fig. 3 a) and thus simulate only weak ozone depletion and consequently weak Arctic ozone minima.
In turn, models with a cold temperature bias (e.g. CCSRNIES-MIROC3.2, red markers) simulate large ClOx concentrations,
strong ozone depletion, and large ozone minima. Models which best reproduce the Arctic mean temperature from MERRA2
generally also agree better with reanalysis in terms of the magnitude of the simulated Arctic ozone minima. This behaviour

compares well with previous results showing that temperature biases limit the models' ability to reproduce observed PSC
coverage (Snels et al., 2019; Steiner et al., 2021).

Besides temperature biases, differences in total inorganic chlorine (Cly) across models could partly be responsible for model
differences in ClOx. A large inter-model spread in Cly in CCM simulations has been reported previously by Eyring et al.
(2006, 2007) and has been attributed to differences in transport of chemical species within the stratosphere (Eyring et al.,

2006). Moreover, differences in the number of chlorine source gases considered (Morgenstern et al., 2017) and in the overall
treatment of photolysis in the models (Sukhodolov et al., 2016) could contribute to the biases in the ozone minima. However,
for the Arctic region, correlation beetween Cly and ClOx concentrations is low across the models analysed here (see Fig. A7).

### 3.3 The source of model differences in future projections

Now that we have established the reasons for differences in simulated ozone minima in current climate across models, we

investigate the reasons for differences in future ozone minima trends. Trends in ozone loss are impacted by both the decline
in hODS concentrations as well as potential changes in stratospheric temperature due to GHG emissions, which might impact
the formation of PSCs. In addition, dynamical changes might contribute to temperature trends, as discussed below. Besides, an
increase in stratospheric water vapor might change the abundance of PSCs in the future (von der Gathen et al., 2021). In Fig. 4
we investigate both the relationship of declining ClOx as well as temperature trends with changes in ozone minima. Trends in

ozone minima are strongly correlated with changes in ClOx concentrations; models with large ClOx concentrations in current



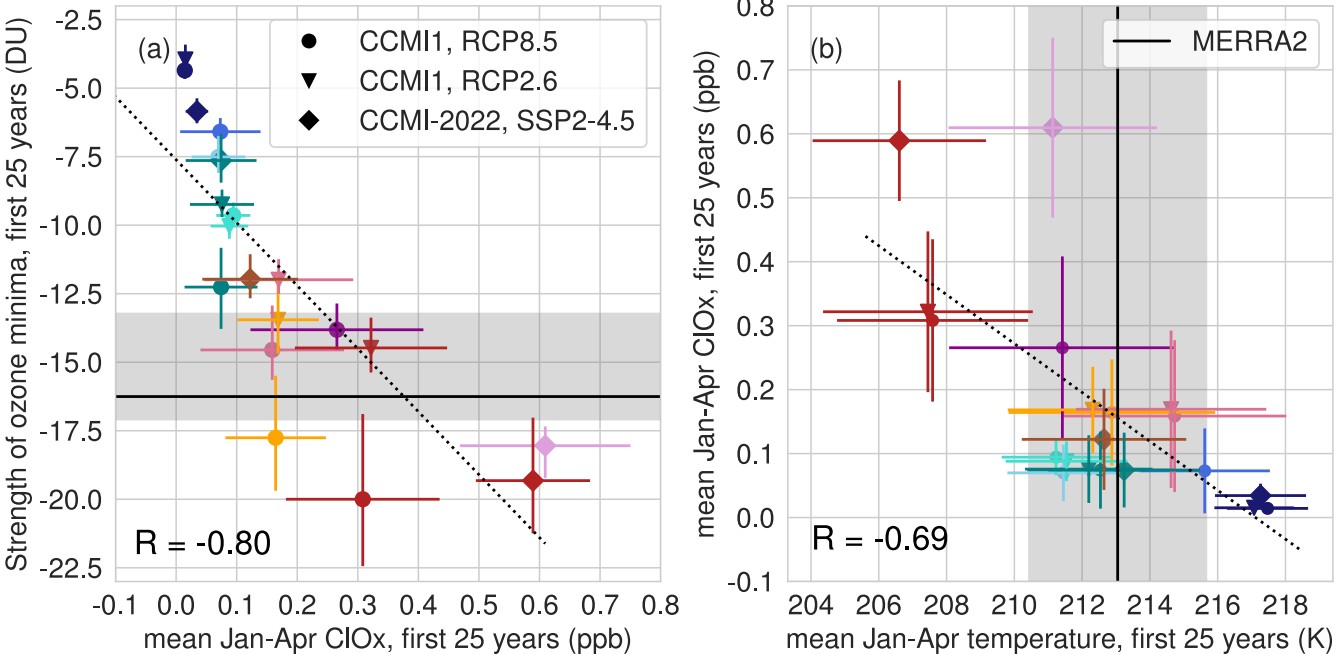

**Figure 3.** Relation of ClOx concentration in late winter/spring (Jan-April) at 50 hPa and the strength of the ozone minima in the first 25 years (a) as well as relation between ClOx concentrations and mean temperature in Jan-April in the first 25 years of simulation (b). Colors indicate the different models as in Figs. 1 and 2. Black dotted lines show the linear regression. The Pearson correlation coefficient is denoted by "R". The mean ozone minima strength as well as the mean temperature in MERRA2 are shown by the black lines. The shading and vertical error bars in (a) show the uncertainty of ozone minima strength, as explained in the methods section. Grey shading as well as error bars in (b) show the standard deviation.

climate (e.g. CCSRNIES-MIROC3.2, red markers) show a large decline in ClOx over the next century and therefore a large decline in ozone loss. Models with little ClOx concentrations to start with (e.g. CMAM, dark blue markers) show almost no changes in active chlorine species in the future, and thus barely any changes in ozone loss. The development of ozone minima in individual CCMs is therefore strongly driven by changes in stratospheric ClOx concentrations.

Next, we investigate the relation between long-term changes in Arctic stratospheric temperature and ozone minima. For ozone minima, the temperature evolution of the coldest winters/springs is most relevant, as large amounts of PSCs and severe ozone loss are expected only under sufficiently cold conditions (< 196 K). Further, it has previously been shown that GHG cooling especially impacts the PSC formation in extremely cold Arctic winters (Rex et al., 2004; Tilmes et al., 2006; von der Gathen et al., 2021). We therefore focus on the temperature evolution of the 20% of coldest winter to spring seasons

(January – April mean) in a 25-year running window, similar to Morgenstern et al. (2022). Overall, CCMs do not agree on the sign of stratospheric temperature trends in late winter/spring in especially cold years (Fig. 4 b). In addition, models also do not agree on the temperature response to additional GHG forcing (see temperature trends for RCP2.6 vs. RCP8.5 in the





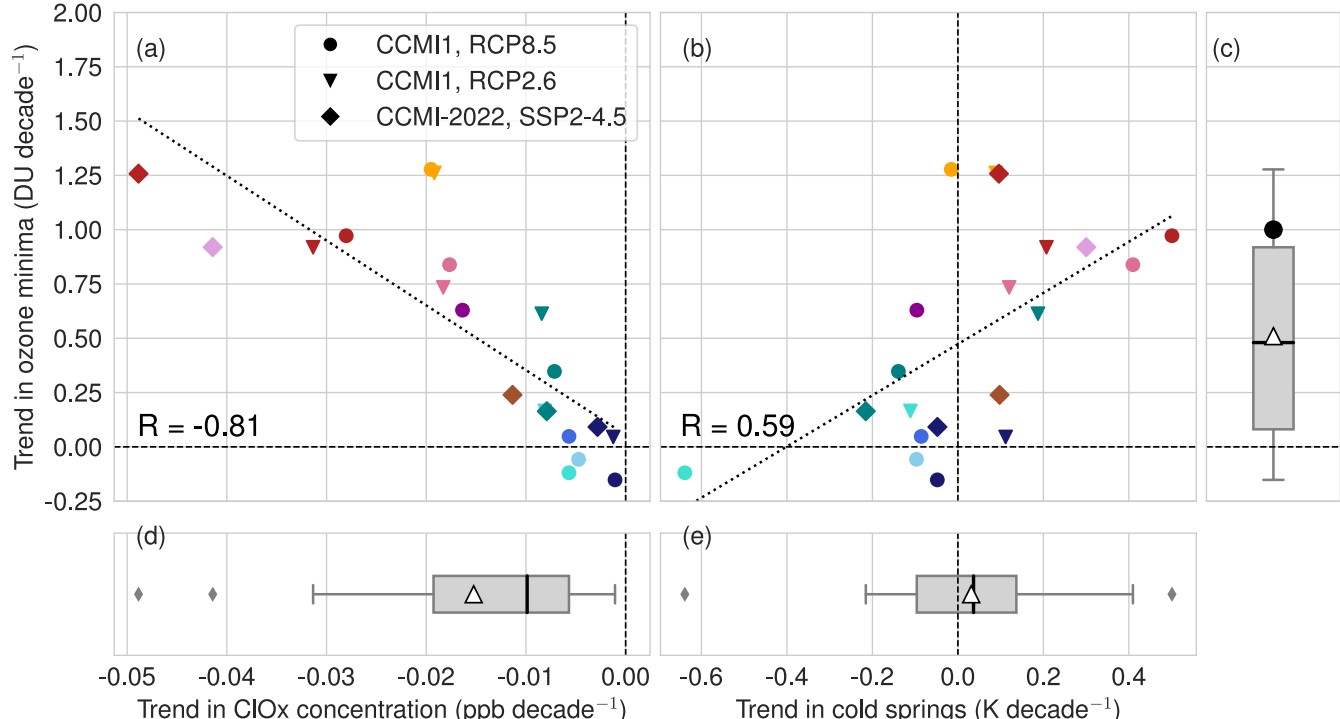

**Figure 4.** Dependence of the trend in ozone minima on the trend in ClOx concentrations (a) as well as on the 50 hPa temperature trend in cold springs (b). Colors indicate the different models as in Figs. 1 and 2. A positive trend in ozone minima strength means a decrease in the magnitude of ozone minima in the future. Negative temperature trends mean that cold winters/springs are getting colder, and positive temperature trends mean that cold winters/springs will be less extreme in the future. Black dotted lines show the linear regression. The Pearson correlation coefficient is denoted by "R". Box plots show changes in the mean ozone minima strength (c), ClOx changes (d) and temperature changes in cold springs (e) across models and scenarios. Triangles mark the median change, black lines the mean change across models. The circle in (c) shows the weighted arithmetic model mean. Note that not all models used to calculate the weighted mean are shown in (a) and (b) due to lack of ClOx data in some models.

individual models). On average, models show a slightly positive temperature trend (consistent with the weak but significant Arctic warming projected in boreal spring reported in the WMO (2018) assessment; see their Fig. 5-8), but the spread ranges

from -0.6 K decade$^{-1}$ to +0.5 K decade$^{-1}$ (Fig. 4 e). This inter-model spread is consistent with what has been reported previously for CCMVal2 models (Bohlinger et al., 2014). Overall, the correlation between stratospheric temperature trends and changes in ozone minima is small (0.59) and mainly caused by the two most extreme negative (ULAQ-CCM) and positive (CCSRNIES-MIROC3.2) values. For the bulk of the models that show no or only minor temperature changes (within -0.2 and 0.2 K decade$^{-1}$), the temperature trends do not seem to be connected with trends in ozone minima (Fig. 4b). Thus, for this ma-

jority of models, different trends in ClOx concentrations are driving different trends in ozone minima, and temperature changes only play a secondary role. However, for models with large temperature trends, like e.g. ULAQ-CCM (-0.6 K decade$^{-1}$) and





CCSRNIES-MIROC3.2 ($+0.5$ K decade$^{-1}$), changes in temperature seem to be reflected in ozone minima trends. For example, in the extreme example of ULAQ-CCM, cold winter/spring seasons are getting extensively colder in the future (see turquoise round marker in Fig. 3 b), which results in a more efficient activation of ClOx. The more efficient activation of ClOx opposes
the decline in atmospheric CFC concentrations. As a result, the ClOx concentration and magnitude of ozone minima hardly changes in this model (see Figs. 2 b and 1).

To highlight the uncertainty in Arctic stratospheric temperature trends in CCMs, we show temperature trends in extremely cold boreal winters (January – April mean) for the whole atmosphere for the high emission scenario RCP8.5 in Fig.5. Although the models agree on the sign of temperature changes in the troposphere and most of the stratosphere, there are large
inter-model differences in the Arctic lower stratosphere (marked by the grey square) with some models projecting a warming, and others projecting cooling. This uncertainty is most likely due to several competing processes that contribute to stratospheric temperature trends over the Arctic; GHGs radiatively cool the stratosphere. This GHG cooling is responsible for the negative temperature trends in large parts of the stratosphere. At the same time, the forthcoming recovery of Arctic ozone (see Fig. A1) is expected to radiatively heat the stratosphere, offsetting a great part of the GHG cooling (Maycock, 2016; Kult-Herdin et al.,
2023). In addition to changes in radiation, changes in large-scale dynamics are expected to impact stratospheric temperature. In particular, a projected strengthening of the Brewer-Dobson circulation (BDC) due to increasing GHGs will drive a stronger downwelling and associated adiabatic dynamical heating over the pole (Butchart, 2014). Since CCMs show different sensitivities to GHG and hODS forcings (Morgenstern et al., 2018), the contribution of the individual processes to temperature trends might vary across models. For example, the evolution of stratospheric dynamics and dynamical variability in the Arctic is very
uncertain and highly model dependent (Ayarzagüena et al., 2018; Ayarzagüena et al., 2020; Abalos et al., 2021; Karpechko et al., 2022), which likely contributes to the uncertainty in Arctic stratospheric temperature trends.

In summary, changes in the magnitude of ozone minima are strongly correlated to the decrease in ClOx across models. Changes in the temperature of cold winters, however, only correlate with changes in ozone minima in models with extreme temperature trends ($> \pm 0.2$ K decade$^{-1}$). Thus, we conclude that long-term changes in Arctic ozone minima are strongly
driven by long-term changes in stratospheric ClOx concentrations. Changes in temperature, on the other hand, seem to play a secondary role in the evolution of Arctic ozone minima for the majority of the models. The relation between changes in ClOx and changes in ozone minima serves as underlying physical mechanism for the emergent constraint analysis as described above. Even if temperature trends are not decisive for the development of ozone minima, it should again be emphasised that temperature biases in the mean state are important to explain the large model scatter in the magnitude of ozone minima under
present-day conditions.

## 4 Conclusions

## 5 Discussion and Outlook

Previous studies reported a large spread in Arctic ozone minima across CCMs and questioned the reliability of simulated ozone (von der Gathen et al., 2021; Morgenstern et al., 2018). Therefore, past studies derived trends in ozone loss based on trends in





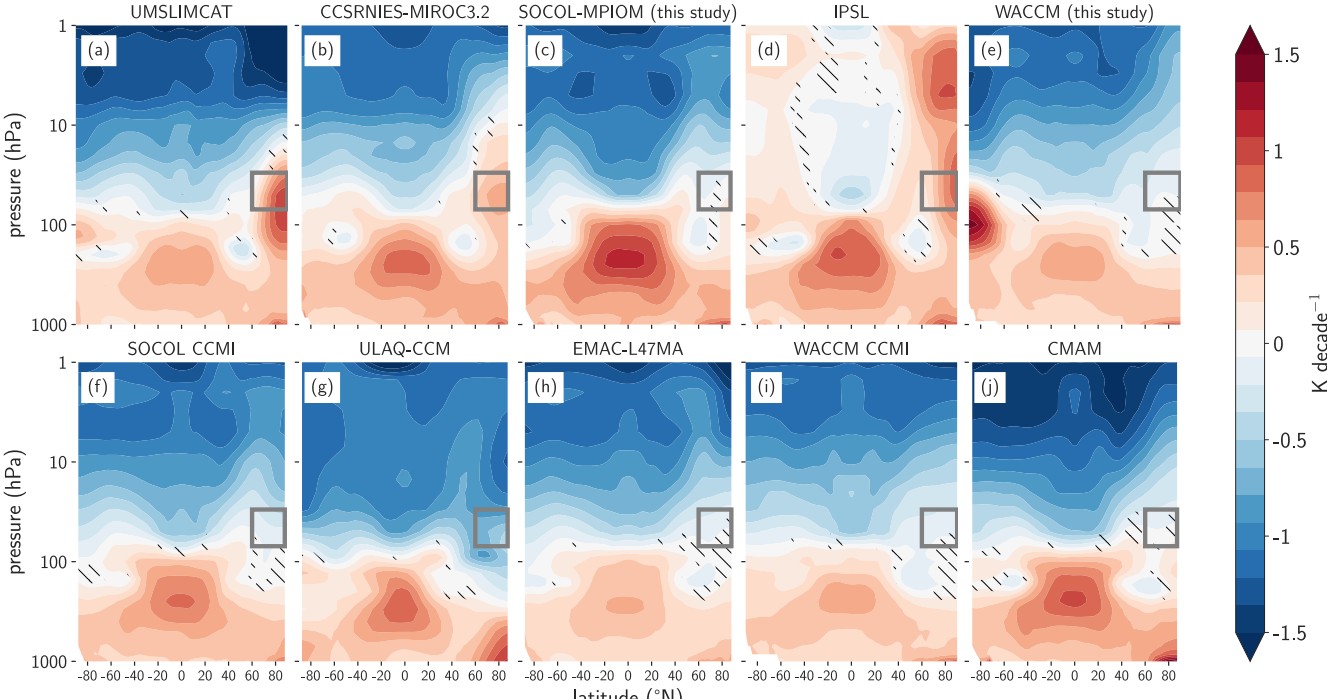

**Figure 5.** Temperature trend of the 20% coldest winters/springs (January – April mean) in a 25-year running window over the course of the 21st century in CCMI1 models for RCP8.5. Stippling marks regions which are not significant on a 5% level. The grey square marks the region of interest (60-90°N polar cap, 30-70 hPa).

temperature and PSC formation potential rather than ozone itself (Rex et al., 2004; Rieder and Polvani, 2013; Langematz et al., 2014; von der Gathen et al., 2021). This study sheds new light on the origin of these model differences, and shows how they can be useful in constraining future projections. Here, we show that differences in the magnitude of ozone minima across models under current conditions are largely due to temperature biases, which lead to different amounts of active chlorine species in the Arctic polar stratosphere. The amount of stratospheric ClOx in the Arctic and thus the magnitude of the ozone minima at

the beginning of the 21st century thereby determines the future trend of negative ozone anomalies: models with high chlorine activation and large ozone minima show a large trend towards less pronounced ozone minima in the future, while models with little chlorine activation and small ozone minima hardly show any trends. Therefore, the uncertainty in the magnitude of ozone minima will decrease in the future, leading to a better agreement of future ozone minima in CCMs. Moreover, the spread across CCMs can be an advantage in constraining the evolution of ozone minima, as the initial strength of the ozone minima

is strongly correlated with its trend. An emergent constraint approach estimates a decline in the magnitude of Arctic ozone minima of about -1 DU decade $^{-1}$, and model simulations suggest that the most severe Arctic ozone anomalies are unlikely to surpass -20 DU by the end of this century. Drastic ozone depletion events, like e.g. the one observed in spring 2020 (Lawrence et al., 2020), will thus become very unlikely by the end of this century. A similar result can be gained when weighting the model



projections according to their performance and interdependence. Such a weighted model average again suggests a decline in
the magnitude of Arctic ozone minima of -1 DU decade $^{-1}$, independent of the GHG scenario.

The absence of extreme Arctic ozone minima past 2070 in the CCMs analysed here stands in an apparent contrast to results
reported by von der Gathen et al. (2021) who suggest that large Arctic ozone loss might still be possible or even increase by
the end of the 21st century under high GHG emission scenarios. However, there are substantial differences in the methods
and variables used in the two studies. First, von der Gathen et al. (2021) infer chemical ozone loss inside the polar vortex
area from temperature trends in CMIP6 models, whereas here we analyse the actual ozone output averaged over the polar
cap from CCMs. As such, the ozone minima analysed here are the result of both, chemical ozone loss and changes in ozone
transport, and thus represent the full extent of the ozone anomaly instead of just the chemical contribution. Second, differences
in the results might arise from different time periods considered: while the results presented here focus on average seasonal
springtime ozone (March – April), von der Gathen et al. (2021) focus on changes in PSC formation potential over the whole
winter to spring period on daily resolution. In addition, von der Gathen et al. (2021) adjust the calculated ozone loss according
to estimated changes in stratospheric water vapor, while in the CCMs presented here such changes are calculated interactively
in the models. Taken together, the results presented here are not necessarily inconsistent with results from von der Gathen et al.
(2021), but rather complement their study by considering the full extent of negative ozone anomalies over the whole season
rather than short-term chemical ozone loss. Similarly, the decrease in Arctic ozone minima as suggested by CCMI models
seems to contradict results from Bednarz et al. (2016) and Akiyoshi et al. (2023), who found that large ozone minima past 2060
might be still be possible in their models (UM-UKCA and MIROC3.2), although rarely. The versions of these models (NIWA-
UKCA2 and CCSRNIES-MIROC3.2) analysed here are both outliers in terms of present-day polar ClOx concentrations (see
Fig. 3 a, pink and red diamonds), which cannot be explained by the models' temperatures (see Fig. 3 b). In these models, there
is still a comparably large amount of ClOx available at the end of the 21st century. While these conditions might be responsible
for the episodic ozone minima past 2060 reported by Bednarz et al. (2016) and Akiyoshi et al. (2023), there is no sign for a
worsening of ozone minima in the future in these models. Rather, the two models consistently indicate a decreasing magnitude
of ozone minima over time.

Ozone minima have previously been reported to influence Northern Hemispheric spring climate via their impact on strato-
spheric temperature and dynamics (Friedel et al., 2022a, b). With the reduction of such ozone minima in future climate, their
ability to influence stratospheric temperatures may diminish, and consequently their role as a driver of springtime surface cli-
mate may become less important. However, there is no consensus on the development of stratosphere-troposphere coupling
in the future, and further investigation is necessary to make conclusion about the relevance of future Arctic ozone minima
for tropospheric climate. Due to the changes in the Arctic mean ozone levels, extreme Arctic ozone minima in the future will
hardly surpass the mean ozone levels of today. Health related impacts of ozone minima (due to impacts on UV exposure) are
therefore likely to decrease. It is to be noted, though, that the results presented here are based on seasonal averages, which
might mask processes on a days to weeks basis that are potentially important for health and climate.

As negative ozone anomalies decrease, so does interannual ozone variability (see Fig. 1). Under current conditions, ozone
variability is an important driver of Arctic stratospheric temperature and dynamical variability in CCMs (Rieder et al., 2019;



Friedel et al., 2022b), and interactive ozone chemistry is considered important for a realistic representation of the stratosphere.
Moreover, the statistical connection between stratospheric ozone anomalies and surface climate suggests that accounting for interactive ozone chemistry in forecast models could provide a potential source of predictability on subseasonal-to-seasonal scales. Whether this relationship and the potential importance of interactive ozone for predictability will hold in the future will require further investigation. However, since not only ozone minima, but also positive ozone anomalies have been shown to significantly impact surface climate in spring (Friedel et al., 2022b), ozone variability can be expected to continue playing a
role for both stratospheric and surface climate in the future.




# Appendix A: Additional Information

## A1 Ozone distributions

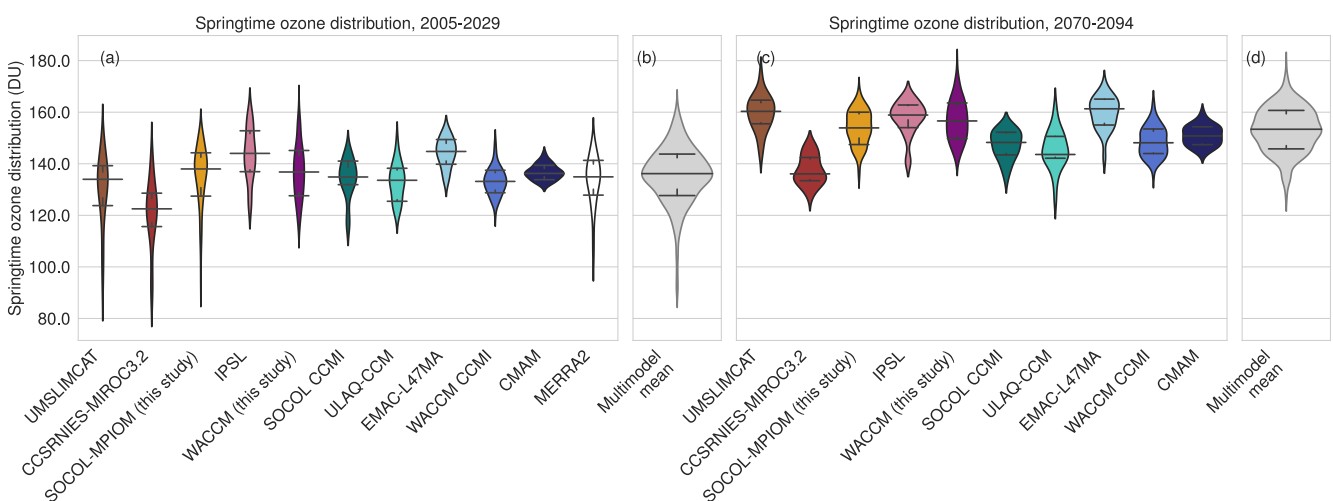

**Figure A1.** Same as Fig. 1 but with absolute ozone values instead of ozone anomalies. As such, the change in distributions between the early (a) and late (b) 21st century convey both changes in ozone extremes (see lower tail of the distributions) as well as the ozone recovery, which is reflected in changes of the climatology.

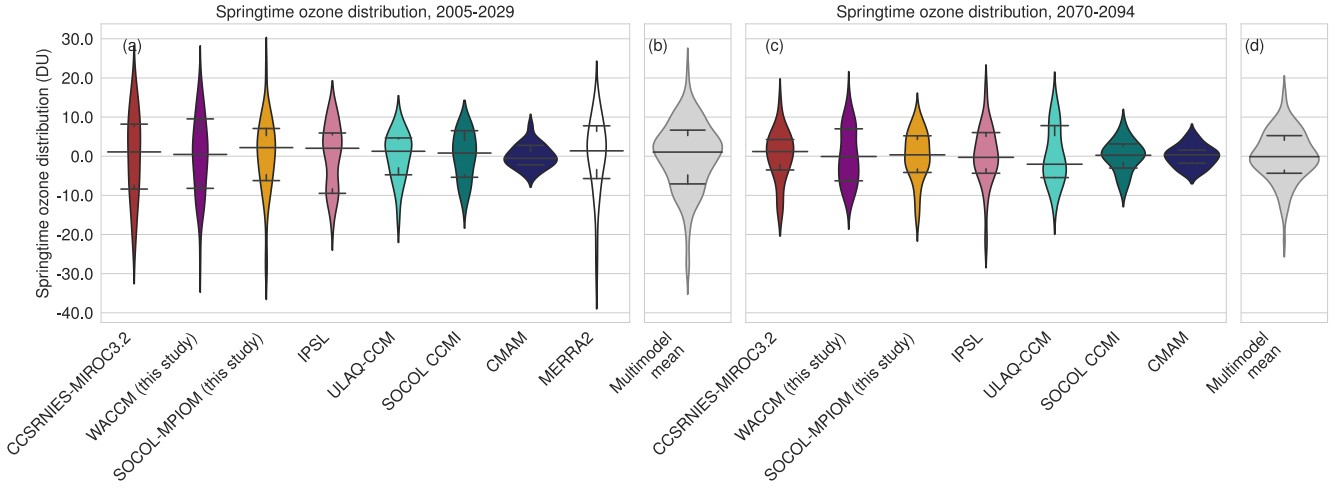

**Figure A2.** Same as Fig. 1 but for CCMI-1 RCP2.6.



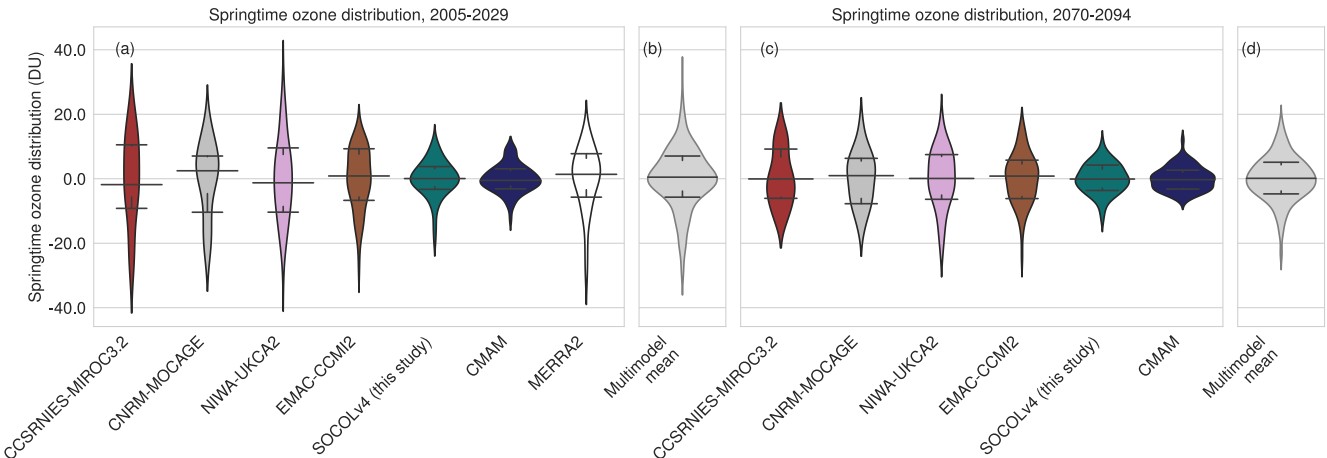

**Figure A3.** Same as Fig. 1 but for CCMI-2022 ref-D2

## A2    Calculation of the weighted model mean

A weighted model average is calculated to estimate the trend in the magnitude of Arctic ozone minima, similar to the method

used by Knutti et al. (2017) and Amos et al. (2020). Model weights are calculated based on their ability to represent the magnitude of ozone minima under present-day conditions. Given $N$ models, the weight $w_i$ of model $i$ is calculated according to

$$w_i = \frac{\exp(D_i^2/\sigma_D^2)}{1 + \sum_{j=1}^{N} \exp(S_{ij}^2/\sigma_S^2)} \tag{A1}$$

where $D_i$ is the difference between the simulated and the observed magnitude of ozone minima, and $S_{ij}$ the difference

between models $i$ and $j$. $\sigma_D$ and $\sigma_S$ are both assumed to be 0.01, as in Amos et al. (2020). Weights are then normalised so that their sum is equal to one (Knutti et al., 2017). The weights calculated following this method are shown in Fig. A4. A weighted arithmetic mean of the trajectories for the ozone minima strength is then calculated for each scenario independently (see Fig. A5), and trends of the mean trajectories are calculated. The trends derived this way are 1.0 (RCP8.5), 1.1 (RCP2.6) and 1.0 (SSP2-4.5) DU decade$^{-1}$.



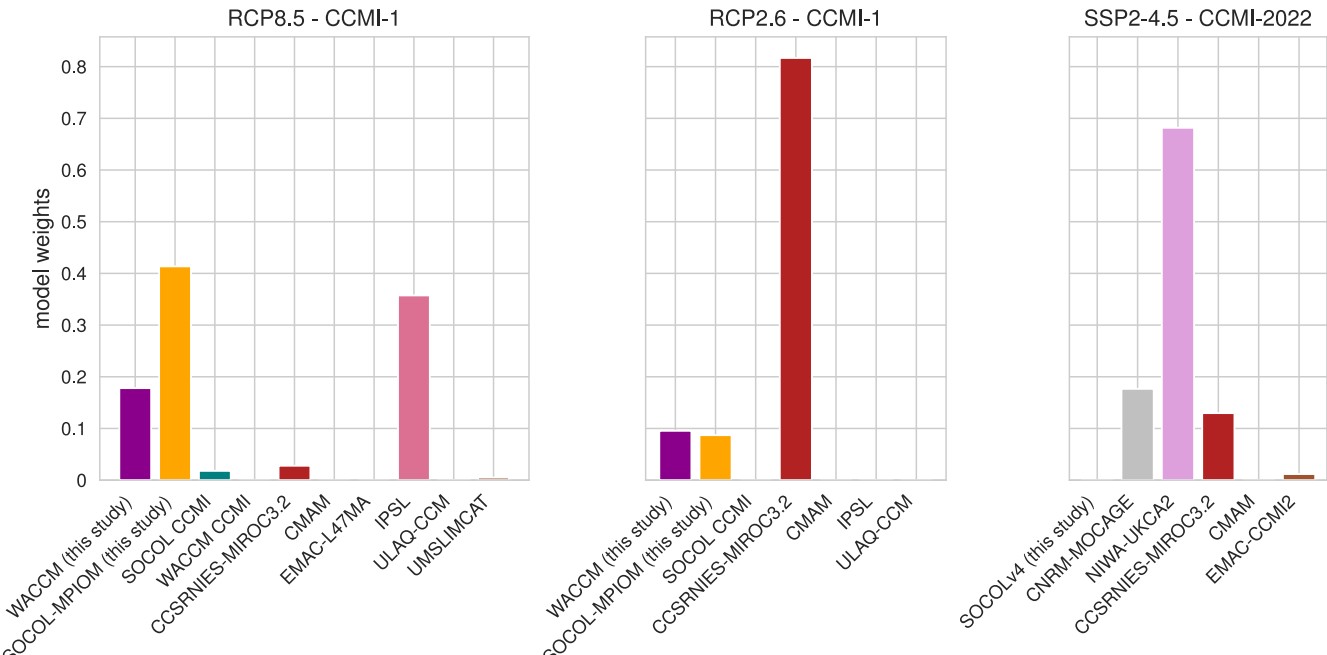

**Figure A4.** Model weights calculated based on the model's ability to reproduce observed ozone minima, as well as interdependence.

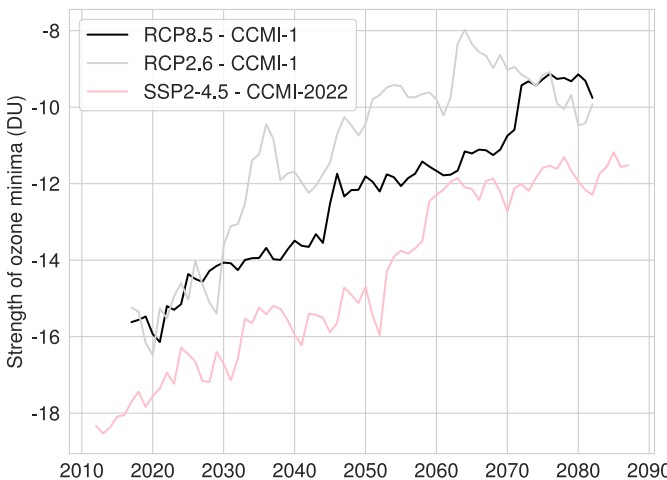

**Figure A5.** The weighted arithmetic mean for the development of the ozone minima strength in the three scenarios considered.





## 370    A3    Timeslice simulations

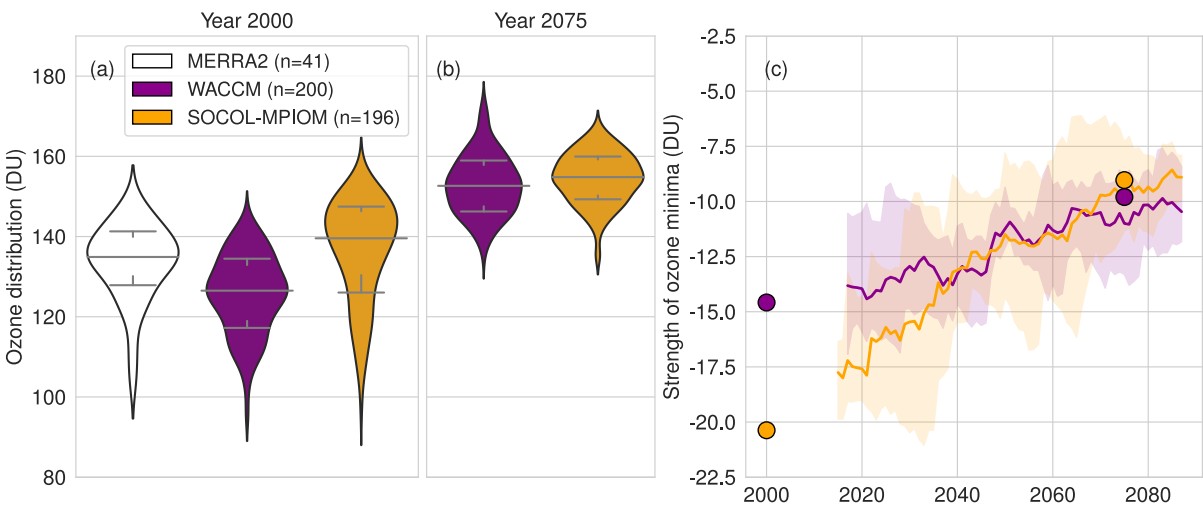

**Figure A6.** Mean distribution of springtime Arctic ozone in timeslice simulations of the year 2000 for SOCOL-MPIOM and WACCM, as well as MERRA2 mean springtime ozone distribu- tion from 1980-2020 (a). Mean distribution of springtime Arctic ozone in timeslice simulations of the year 2075 for SOCOL-MPIOM and WACCM (b). Development of the strength of ozone minima in WACCM and SOCOL-MPIOM RCP8.5 simulations (solid lines) as well as the mean strength of the 20% strongest ozone minima in the timeslice simulations for the years 2000 and 2075 (circles). Shading shows the maximum and minimum values across the 5 ensemble members.

## A4    Relation of Cly and ClOx in CCMI-1 RCP8.5

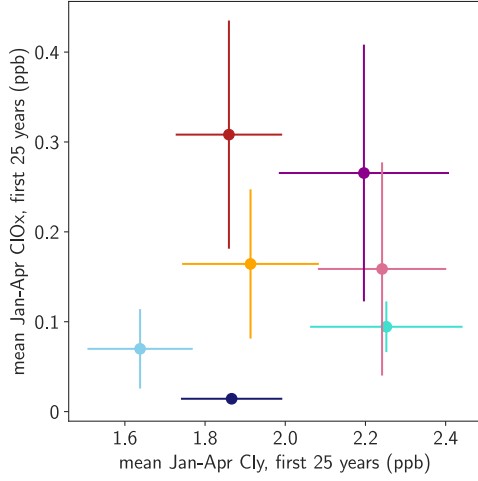

**Figure A7.** Relation of Cly and ClOx concentration (2005-2029 climatologies) in late winter/spring (Jan-April) at 50 hPa for CCMI-1 models under RCP8.5. Colors indicate the different models as in Fig. A1.



*Code and data availability.* The CCMI-1 and CCMI-2022 data used in this study can be obtained through the British Atmospheric Data Centre (BADC) archive (http://data.ceda.ac.uk/badc/wcrp-ccmi/data/CCMI-1/output/ and https://data.ceda.ac.uk/badc/ccmi/data/post-cmip6/ccmi-2022). The present-day timeslice simulations used in this study are available in the ETH Research Collection. Data for WACCM: https://www.research-collection.ethz.ch/handle/20.500.11850/527155(Friedel and Chiodo, 2022b). Data for SOCOL-MPIOM: https://www.research-collection.ethz.ch/handle/20.500.11850/546039 (Friedel and Chiodo, 2022a). Corresponding data for the timeslice simulations of the year 2075 and transient simulations for WACCM and SOCOL-MPIOM, as well as all scripts used for the analysis in this study are available upon request. The MERRA2 reanalysis data can be downloaded from the Goddard Earth Sciences Data and Information Services Center (GES DIC) (https://disc.gsfc.nasa.gov/datasets?keywords=%22MERRA-2%22&page=1&source=Models%2FAnalyses%20MERRA-2).

*Author contributions.* M.F., G.C., T.S., A.S. and S.S. performed and processed the SOCOL and WACCM experiments. H.A., E.R., D.P., P.J., G.Z., O.M. and B.J. performed and processed the CCMI experiments. M.F. analysed the results, M.F., G.C., T.P. and J.K. interpreted the results. M.F. wrote the paper with input from all authors.

*Competing interests.* The authors declare that they have no conflict of interest.

*Acknowledgements.* We acknowledge the modeling groups for making their simulations available for this analysis, the joint WCRP SPARC/IGAC Chemistry-Climate Model Initiative (CCMI) for organizing and coordinating the model data analysis activity, and the Centre for Environmental Data Analysis (CEDA) for collecting and archiving the CCMI model output. Support from the Swiss National Science Foundation through Ambizione Grant PZ00P2_180043 for M.F. and G.C. is gratefully acknowledged. The EMAC model simulations have been performed at the German Climate Computing Centre (DKRZ) through support from the Bundesministerium für Bildung und Forschung (BMBF). DKRZ and its scientific steering committee are gratefully acknowledged for providing the HPC and data archiving resources for this consortial project ESCiMo (Earth System Chemistry integrated Modelling). J. K. would like to thank the Met Office CSSP-China Programme for providing funding support through the POzSUM project and NERC for funding through the InHALE project. H. A. acknowledges Environment Research and Technology Development Fund of the Environmental Restoration and Conservation Agency, Japan (2-1303 and JPMEERF20172009), KAKENHI (JP18KK0289 and JP20H01977) of the Ministry of Education, Culture, Sports, Science, and Technology, Japan, and NEC SX-ACE and SX-AURORA TSUBASA computers at NIES. The authors wish to acknowledge the use of New Zealand eScience Infrastructure (NeSI) high performance computing facilities, consulting support and/or training services as part of this research. New Zealand's national facilities are provided by NeSI and funded jointly by NeSI's collaborator institutions and through the Ministry of Business, Innovation & Employment's Research Infrastructure programme. URL https://www.nesi.org.nz. G.Z. and O.M. acknowledge funding by the New Zealand Ministry of Business, Innovation and Employment (MBIE) under their Strategic Science Investment Fund (SSIF). E.R. and T.S. acknowledge support from the Swiss National Science Foundation (grant 200020-182239). Calculations with the SOCOLv4 were performed at the Swiss National Supercomputing Centre (CSCS) under projects S-901 (ID 154), S-1029 (ID 249), and S-903.



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
