# Peer review of "Weakening of springtime Arctic ozone depletion with climate change"

_EGUsphere, 2023_

## Author Comment (AC1)

**Reply to reviews: Weakening of springtime Arctic ozone depletion with climate change**

Marina Friedel, Gabriel Chiodo, Timofei Sukhodolov, James Keeble, Thomas Peter, Svenja Seeber, Andrea Stenke, Hideharu Akiyoshi, Eugene Rozanov, David Plummer, Patrick Jöckel, Guang Zeng, Olaf Morgenstern, Béatrice Josse

RC = Reviewer Comment

AR = Author Reply

**Reviewer 1**

We thank the reviewer for their comments, which helped us to clarify some aspects of our manuscript. Please find the point-by-point response below.

**RC 1.1**

Overview:

This paper addresses the issue whether the combination of decreasing halogenated ozone-depleting substances and increasing greenhouse gases will lead to changes in the frequency of major Arctic ozone minima by the end of the 21st century. Complicating the matter is that there are large differences in CCMI (1&2) models, not all of which are able to simulate the magnitude of Artic ozone minima in the current climate. They find that models that overpredict ozone minima in the present climate (those with a cold pole bias) have a decrease in the number of ozone minima under a future GHG scenario. Those that are warm biased have small sensitivity to changes in both GHGs and ODS concentrations. Overall, they find the Arctic ozone minima will lessen with increases in GHGs, largely due to decreases in ODSs. The stratospheric cooling caused by increases in GHGs that could potentially increase ozone minima is weakened by opposing radiative and dynamical mechanisms.

It is found that models give different answers because they have different sensitivities to GHG and ODS forcings, in particular with respect to lower stratospheric transport and dynamics, and thus have a large inter-model spread in temp and ozone trends. To come to their conclusions, given all the different model results, the authors compared the ozone evolution across different CCMs and GHG emission scenarios, while identifying reasons for model discrepancies. They then linked the model spread in future ozone trends to differences in model climatologies, and compared with observations to identify the likely evolution of future Arctic ozone minima.

This is a well written manuscript covering a timely topic. How Artic ozone will respond to increasing greenhouse gases and decreasing ozone depleting substances is of interest to the parties of the Montreal Protocol and the stratospheric ozone community. I recommend publication after considering the comments below.

Page 2, line 33; this references WMO 2018... the authors may want to check whether the statement is still supported by WMO 2022.

**AR 1.1**

Thank you, this is a good point. We checked and this statement is still supported by the WMO 2022 ozone assessment:

*"No statistically significant signature of recovery in Arctic stratospheric ozone over the 2000–2021 period has yet been detected. Observed Arctic ozone trends remain small compared to the year-to-year dynamical variability."* (page 221, scientific summary of chapter 4)

We changed the reference accordingly.

**RC 1.2**

Page 3, line 65, add a comma after "two CCMs" and after "version 4"

**AR 1.2**

This has been adapted.

**RC 1.3**

Question, why do you normalize to the period mean ozone? From a surface UV point of view, it's the absolute ozone value rather than a deviation from average that matters. So, it seems one should identify the ozone minima based on an absolute DU threshold rather than a deviation from a period climatology. At a minimum, in Figure 1, I'd include a notation of what the mean used is. And, it seems that the horizontal scale for the multimodel means (b&d) is different from the individual models, and for ease in looking at the figure, it should be the same.

**AR 1.3**

There is large scientific consensus on the projections of Arctic stratospheric mean ozone, which is projected to increase in all emission scenarios until the end of this century (WMO, 2022, see e.g. Fig. 4-24). However, there are large uncertainties about how deviations from the mean ozone, especially negative deviations following ozone depletion, will develop in the future, as discussion around the study by von der Gathen et al. (2021) shows (Polvani et al., 2023; von der Gathen et al., 2023).

In order to shed new light on this discussion and find reasons for differing findings across studies, this paper focuses on negative deviations from the mean rather than absolute changes in ozone. We further show in Fig. A1 that those changes in

[Figure]

**Figure R1.** Same as Fig. 2 a,c,e, but for absolute values of ozone minima.

50  ozone minima are accompanied by an overall increase in mean ozone in all scenarios. We therefore show that not only the mean Arctic ozone abundance increases, but also that negative deviations from the mean are getting smaller. In addition, one would have to look at total column ozone instead of stratospheric ozone column to get the full picture of UV exposure.

However, following this comment, we have reproduced Fig. 2 (panels a, c, e) in Fig. R1 for absolute changes in ozone minima (defined as the lowest quartile of absolute springtime ozone values in a 25-year running window). The absolute strength of
55  ozone minima decreases in all scenarios and models, with the overall increase being scenario dependent.

**RC 1.4**

This is related to the normalization: on page 7 line 165, you note looking at ozone minima in a 25-year running window. What do you use for the base ozone for identifying the extreme ozone minima events? Is it still one value for the start of the time serios, and a second for the end? Are there less events at the end of the time series? (shown in Figure 2)

60  **AR 1.4**

We are sorry that the methodology was apparently not explained in sufficient detail. Ozone minima are always defined as an average over the 5 strongest negative ozone anomalies with respect to the ozone climatology in the corresponding window. As such, the underlying ozone climatology which we use to calculate ozone anomalies is constantly changing over the century. Reason for this is that we are interested in the maximum possible interannual ozone deviations rather than mean ozone changes
65  (as explained above). We have now added a more detailed explanation of the methodology near line 167. Since we define ozone minima relative to the changing climatology and do not use a fixed threshold, every running window has the same amount of ozone minima per definition. We agree that the frequency with respect to a fixed threshold is likely decreasing over the course

of the century, which would also be reflected in the decreasing amplitude of the ozone minima strength as shown in Fig. 2.

70  **RC 1.5**

Page 8, line 210:should say a reduction in the strength of the BDC.

**AR 1.5**

This has been changed to: "weakening of the BDC".

**RC 1.6**

75  Page 10, line 229: Rather than saying agree better with reanalysis (which is rather generic) I recommend saying agree better with MERRA 2 ozone if that's what you compared with. Would be even better to compare ozone with satellite measurements (such as MLS).

**AR 1.6**

The wording has been changed accordingly. We originally only showed MERRA2 data, since the reanalysis record is longer
80  compared to satellite products (MLS is only available starting from 2004). However, we now added data of the SWOOSH ozone dataset in Figs. 2 and 3. Due to missing data, we decided to only use data from 2004 onward and calculate the ozone minima strength in SWOOSH based on the 3 strongest ozone minima in 2004–2020 ($20^{th}$ percentile).

**RC 1.7**

Page 13, line 283 says "the contribution of individual processes to temperature trends might vary across models" It seems that,
85  with the model output you have, you can show this. Take 2 different models with different trends and analyze the output to see what's driving the temperatures trends.

**AR 1.7**

Thank you for this comment. We have considered this suggestion, and think that with the available data we will not be able to explain causes for varying temperature trends across models. While we could establish correlations between temperature and
90  changes in chemical composition/dynamics, temperature changes are not easily attributable to such individual processes. In other words, a causal connection cannot be established with the given simulations, without running idealized experiments that isolate the role of each process.

Bohlinger et al. (2014) analyzed the spread in lower stratospheric temperature projections across CCMVal models using regression analysis. They found that the spread in temperature projections is both driven by radiative (i.e. radiatively active gases)

 and dynamical (i.e. planetary wave activity) changes, suggesting different model sensitivities to both radiative and dynamical processes. A statement on this has been added to the manuscript around line 288.

**RC 1.8**

Page 13, conclusions... Does von der Gathen 2021 really question the reliability of simulated ozone? They really don't even use simulated ozone, instead they use assorted proxies. And after rereading Morgenstern et al., I don't think it supports questions the reliability of simulated ozone.

**AR 1.8**

Thank you for this comment. You are right, the reliability of simulated ozone is rather questioned by von der Gathen et al. (2023) in the following section:

*"..., it is clear that many of the earlier models with interactive chemistry analyzed by Dhomse et al. and the CMIP6 models analyzed by P22 have difficulty representing observed TCO, ultimately due to deficiency in the model representation of chemical ozone loss. [...] Figure 11 of ref. 12 shows stark differences between observed and simulated TCO in the Arctic polar cap for March, with models failing to capture steep lows observed during particularly cold winters."*

Therefore, we adapted the reference accordingly. Here, we show that only a fraction of the CCMI models fails to reproduce severe ozone loss as observed in recent years. Further, we find reasons for the model discrepancies and show how the model spread can be useful in constraining future projections.

**RC 1.9**

This recent paper is relevant to this study: http://www.columbia.edu/~lmp/paps/polvani+etal-NATURECOMM-2023.pdf (Polvani et al., No evidence of worsening Arctic springtime ozone losses over the 21st century ) One of their conclusions is "When all the relevant process are included, as they are in the state-of-the-art comprehensive chemistry-climate models, there is no evidence that future ozone levels will decrease in the coming decades, including over the Arctic in springtime, as we now explicitly show." This seems to be very similar to the conclusions of this study. This should be refereced around page 15, line 316.

**AR 1.9**

Thank you for this comment. Indeed, Polvani et al. (2023) come to very similar conclusions. Please note at the time this paper has been published our manuscript had already been submitted. We now include reference and discussion of Polvani et al. (2023) in our manuscript in the introduction (near line 55) and in the conclusions (near line 316).

**References**

Bohlinger, P., Sinnhuber, B. M., Ruhnke, R., and Kirner, O.: Radiative and dynamical contributions to past and future Arctic stratospheric temperature trends, Atmospheric Chemistry and Physics, 14, 1679–1688, https://doi.org/10.5194/acp-14-1679-2014, 2014.

125 Polvani, L. M., Keeble, J., Banerjee, A., and et al.: No evidence of worsening Arctic springtime ozone losses over the 21st century, Nature Communications, 14, 1608, https://doi.org/10.1038/s41467-023-37134-3, 2023.

von der Gathen, P., Kivi, R., Wohltmann, I., Salawitch, R. J., and Rex, M.: Climate change favours large seasonal loss of Arctic ozone, Nature Communications, 12, https://doi.org/10.1038/s41467-021-24089-6, 2021.

von der Gathen, P., Kivi, R., Wohltmann, I., and et al.: Reply to: No evidence of worsening Arctic springtime ozone losses over the 21st century, Nature Communications, 14, 1609, https://doi.org/10.1038/s41467-023-37135-2, 2023.

130 WMO, W. M. O.: Scientific Assessment of Ozone Depletion: 2022, GAW Report No. 278, p. 509 pp., 2022.

---

## Author Comment (AC2)

**Reply to reviews: Weakening of springtime Arctic ozone depletion with climate change**

Marina Friedel, Gabriel Chiodo, Timofei Sukhodolov, James Keeble, Thomas Peter, Svenja Seeber, Andrea Stenke, Hideharu Akiyoshi, Eugene Rozanov, David Plummer, Patrick Jöckel, Guang Zeng, Olaf Morgenstern, Béatrice Josse

RC = Reviewer Comment

AR = Author Reply

**Reviewer 2**

We thank the reviewer for their detailed assessment of our manuscript and helpful comments. Please find the point-by-point response below.

**RC 2.1**

Friedel et al. provide a comprehensive study on the future evolution of ozone minima over the Arctic polar cap. The authors present a thorough analysis of a series of transient and timeslice simulations with 2 CCMs (WACCM and SOCOL) as well as the cohort of simulations available from CCMI1 and CCMI-2022 models. In their study the authors: i) address individual model weaknesses (warm and cold biases) related to the realization of ozone minima; ii) explore the related spread in modelled springtime ozone anomalies at present and for different future climate scenarios; iii) quantify the magnitude of ozone anomalies for early and late 21st century and temporal changes in these anomalies; iv) illustrate how the amount of ClOx available in CCMs drives ozone minima at present and also determines their future trends; and v) detail how inter-model spread can be used to constrain ozone minima projections.

The manuscript is timely and well prepared. The study sheds new light on the long-standing question regarding the future evolution of Arctic low ozone extremes and emphasizes the central role of declining ODS abundances for future Arctic ozone, both mean and extreme, across potential climate pathways.

I recommend accepting this manuscript for publication after addressing the comments provided below.

General comments:

Section 3.1, L163-170: The authors examine ozone minima present and future, and the model spread in ozone minima across CCMs. While the text and Fig. 2 illustrate well the overall change in ozone anomalies I would suggest also including a short

text passage or supplemental table detailing the timing of the most pronounced ozone minima per CCM and scenario (on decadal basis).

**AR 2.1**

It is not entirely clear to us what the reviewer is suggesting. Is the seasonal timing of the ozone minima in each spring meant, or rather the timing of the e.g. 5 most pronounced ozone minima in each CCM over the $21^{st}$ century? If it is the former, we are afraid that the calculation of the date of the ozone minima within each season would require data on daily resolution, which is only available for a fraction (ca. 50%) of the models. If the latter is meant, we are unsure what the benefit of such additional information would be. We would therefore like to ask the reviewer to please specify what they mean with "timing of the most pronounced ozone minima" and what the benefit of showing it would be.

**RC 2.2**

Section 3.3, L210-228: I would assume the picture would not change strongly but how would Fig. 3 look like if you restrict to ClOx and temperature in spring (March-April)?

**AR 2.2**

Thank you for this comment. Figure R2 shows the connection between springtime Arctic ozone minima and ClOx/temperature at 50 hPa when only values in March-April are being considered. As one can see, the correlations are still moderate to strong, but weaker than when considering January-April means for ClOx and temperature as in Fig. 3. We believe that when averaging over March-April, a large fraction of the averaging window might be after the chemical ozone depletion (depending on the model). In SOCOL and WACCM, for which we conducted extensive analyses on the seasonal timing of ozone minima, we found that ozone minima typically occur between mid-March and mid-April in these models (Friedel et al., 2022a, b). ClOx and temperature anomalies happening after the ozone minima are no good proxies for the amount of ozone depleted. Rather, one should consider temperature and ClOx anomalies anticipating the ozone minima. In fact, using January-February means for ClOx and temperature yields correlations of -0.80 and -0.69, respectively, which is the same as in Fig. 3 in the paper (where Jan-Apr mean values where chosen for ClOx and temperature).

**RC 2.3**

Section 3.3, L261-264: I would call the correlation between stratospheric temperature trends and ozone minima of 0.59 moderate not weak. However, I agree with the dominance of some models for overall R. Thus, I would recommend specifying how R changes if the two most extreme (positive and negative) models are removed.

[Figure]

**Figure R1.** Same as Fig. 3 in the manuscript, but for March-April averages for ClOx and temperature.

**AR 2.3**

The wording has been changed to "moderate". When calculating the correlation coefficient excluding the two most extreme positive and negative values, R reduces to 0.5.

**RC 2.4**

Section A2: I agree with the authors to apply model weighting, however given the substantial difference in weight for different models across scenarios I would suggest adding a second panel to Fig. A5 showing also the evolution of the unweighted multi-model mean of ozone minima strength.

**AR 2.4**

Thank you for this suggestion. We included the unweighted multi-model mean as stippled lines in Fig. A5.

**RC 2.5**

Technical comments:

L 155: on a multimodel mean → on the multimodel mean

[Figure]

**Figure R2.** Same as Fig. 3 in the manuscript, but for January-February averages for ClOx and temperature.

**AR 2.5**

This has been adapted.

65  **AR 2.6**

This has been adapted.

**AR 2.7**

70  This has been adapted.

**AR 2.8**

This has been corrected.

**RC 2.9**

Figure 1, caption: normalized by mean ozone → normalized by mean partial column ozone

**AR 2.9**

This has been adapted.

**RC 2.10**

Fig. 2, caption: normalized by ozone climatology → normalized by the ozone climatology

**AR 2.10**

This has been adapted.

**RC 2.11**

Fig. A7: is the vertical whisker for CMAM ClOx missing?

**AR 2.11**

Thank you for this comment. The whisker is actually too small to be seen on this plot (the standard deviation for January-April ClOx in CMAM is 0.004 ppmv). Figure R3 shows the Jan-Apr mean ClOx development over the $21^{st}$ century in the lower stratosphere (50 hPa). The dark blue line shows CMAM. It can be seen that CMAM has (i) very little ClOx in the first place, and (ii) the interannual ClOx variability in this model is extremely small. Hence, the uncertainty in this variable for this model is small.

[Figure]

**Figure R3.** Ensemble mean ClOx develoment (Jan-Apr) at 50 hPa over the $21^{st}$ century.

**References**

Friedel, M., Chiodo, G., Stenke, A., Domeisen, D. I., Fueglistaler, S., Anet, J., and Peter, T.: Springtime Arctic ozone depletion forces Northern Hemisphere climate anomalies, Nature Geoscience, 15, 541–547, https://doi.org/https://doi.org/10.1038/s41561-022-00974-7, 2022a.

95 Friedel, M., Chiodo, G., Stenke, A., Domeisen, D. I. V., and Peter, T.: Effects of Arctic ozone on the stratospheric spring onset and its surface impact, Atmospheric Chemistry and Physics, 22, 13 997–14 017, https://doi.org/10.5194/acp-22-13997-2022, 2022b.